# Tracking the incidence and risk factors for SARS-CoV-2 infection using historical maternal booking serum samples

Edward Mullins[1,2☯]*, Ruth McCabe[3,4☯], Sheila M. Bird[5,6], Paul Randell[7], Marcus J. Pond[7], Lesley Regan[1,8], Eleanor Parker[9], Myra McClure[9], Christl A. Donnelly[3,4,10]

1 Department of Metabolism, Digestion and Reproduction, Imperial College London, London, United Kingdom, 2 The George Institute for Global Health, Queen Charlotte's and Chelsea Hospital, London, United Kingdom, 3 Department of Statistics, University of Oxford, Oxford, United Kingdom, 4 NIHR Health Research Protection Unit in Emerging and Zoonotic Infections, University of Liverpool, Liverpool, United Kingdom, 5 MRC Biostatistics Unit, University of Cambridge, Cambridge, United Kingdom, 6 College of Medicine and Veterinary Medicine, University of Edinburgh, Edinburgh, United Kingdom, 7 Department of Infection and Immunity, North West London Pathology, London, United Kingdom, 8 Department of Obstetrics & Gynaecology, St Mary's Hospital, London, United Kingdom, 9 Department of Infectious Disease, Faculty of Medicine, Imperial College London, London, United Kingdom, 10 MRC Centre for Global Infectious Disease Analysis, Department of Infectious Disease Epidemiology, Imperial College London, London, United Kingdom

☯ These authors contributed equally to this work.
* edward.mullins@imperial.ac.uk

**Data Availability Statement:** Data and code to replicate the analysis are available on GitHub (https://github.com/ruthmccabe/historic_antenatal_serostudy).

## Abstract

The early transmission dynamics of SARS-CoV-2 in the UK are unknown but their investigation is critical to aid future pandemic planning. We tested over 11,000 anonymised, stored historic antenatal serum samples, given at two north-west London NHS trusts in 2019 and 2020, for total antibody to SARS-CoV-2 receptor binding domain (anti-RBD). Estimated prevalence of seroreactivity increased from 1% prior to mid-February 2020 to 17% in September 2020. Our results show higher prevalence of seroreactivity to SARS-CoV-2 in younger, non-white ethnicity, and more deprived groups. We found no significant interaction between the effects of ethnicity and deprivation. Derived from prevalence, the estimated incidence of seroreactivity reflects the trends observed in daily hospitalisations and deaths in London that followed 10 and 13 days later, respectively. We quantified community transmission of SARS-CoV-2 in London, which peaked in late March / early April 2020 with no evidence of community transmission until after January 2020. Our study was not able to determine the date of introduction of the SARS-CoV-2 virus but demonstrates the value of stored antenatal serum samples as a resource for serosurveillance during future outbreaks.

## Background

The highly transmissible severe acute respiratory syndrome coronavirus-2 (SARS-CoV-2) causes coronavirus disease 2019 (COVID-19), which has led to widespread suffering and loss of life since its discovery in Wuhan, China in late 2019. In the UK, the first two polymerase chain reaction (PCR)-confirmed COVID-19 cases were confirmed on 2 February 2020 and the

**Funding:** This research was partly funded by Community Jameel and the Imperial President's Excellence Fund. https://www.imperial.ac.uk/jameel-institute/ EM was funded by an NIHR Academic Clinical Lecturer post until February 2021. https://www.nihr.ac.uk/explore-nihr/academy-programmes/integrated-academic-training.htm RM and CAD acknowledge the NIHR HPRU in Emerging and Zoonotic Infections, a partnership between Public Health England (PHE), University of Oxford, University of Liverpool and Liverpool School of Tropical Medicine [grant number NIHR200907]. CAD also acknowledges the MRC Centre for Global Infectious Disease Analysis [grant number MR/R015600/1], which is jointly funded by the UK Medical Research Council (MRC) and the UK Foreign, Commonwealth & Development Office (FCDO), under the MRC/FCDO Concordat agreement and is also part of the EDCTP2 programme supported by the European Union (EU). EP, MM and RST acknowledge support from NIHR CV220-111: Serological detection of past SARS-CoV-2 infection by non-invasive sampling for field epidemiology and quantitative antibody detection and from departmental funds. Infrastructure support was provided by the NIHR Imperial College Biomedical Research Centre. The funders had no role in study design, data collection and analysis, decision to publish, or preparation of the manuscript. There was no additional external funding received for this study.

**Competing interests:** EM is an academic editor at PLOS One. MM is listed as an inventor in IPR filings for the Imperial Hybrid DABA used in this analysis. Please see United Kingdom Patent Application No. 2011047.4 for "SARS-CoV-2 antibody detection assay". SMB is member of both Royal Statistical Society's COVID-19 Taskforce and Working Group on Diagnostic Tests. SMB serves on UKHSA/DHSC's Testing Initiatives Evaluation Board (January 2021 to present). All other authors declare no Conflicts of Interest. This does not alter our adherence to PLOS ONE policies on sharing data and materials.

first official COVID-19 death occurred on 2 March 2020 [1]. However, infection by SARS-CoV-2 does not invariably lead to symptomatic COVID-19: mild or asymptomatic infection is also common [2].

For several reasons, epidemiological surveillance systems based on reported clinical disease typically capture only a subset of the true number of infections and deaths attributable to a specific pathogen. Reasons include, but are not limited to, the occurrence of asymptomatic infections, limited testing capacity and delays in registration of deaths or mis-attributed cause [3]. For example, in England there are earlier deaths attributable to COVID-19 by autopsy, the (known) first of which occurred on 30 January 2020 [4].

Anonymised serological studies are an important mechanism to infer population-level prevalence and incidence of infection and avoid under-reporting inherent in the reliance upon the incidence of clinical infection [5]. The insights gained from such data allow for better understanding of the transmission dynamics of, and demographic groups most at risk from, a particular virus. Linked to self-completed behavioural risks questionnaire, nationwide surveillance surveys to detect SARS-CoV-2 antibodies in England did not start until after the peak of the first wave [6, 7].

Ascertaining estimates of the prevalence and incidence of infection in the early weeks of SARS-CoV-2 transmission are critical learning to inform preparations for future outbreaks. Nonetheless, such studies are challenging to conduct retrospectively because they rely on the availability of routinely stored blood or other biological samples from well-defined sub-populations, such as blood donors [8–10] or pregnant women [11], but typically lack non-demographic risk-factor information.

In the UK, serum samples routinely given by women at booking for antenatal care, and known as 'booking samples', are stored for at least two years post-collection as the samples may be informative in monitoring the pregnancy or diagnosing illness in the infant during the first year of life [12, 13]. Archives of these booking samples present a unique opportunity to test retrospectively for total antibody to SARS-CoV-2 receptor binding domain (anti-RBD), and, by extension, shed light on early transmission dynamics, within a young (18–44 years old), broadly healthy, and ethnically diverse female population across all deciles the Index of Multiple Deprivation (IMD) [14].

In this study, we present a retrospective analysis of antenatal serum samples from two north-west London trusts across October 2019 –September 2020 which were subsequently tested for SARS-CoV-2 seroreactivity. We used the results to estimate the prevalence of seroreactivity over time and in association with age, ethnicity, and deprivation and to examine the trends in incidence of seroreactivity over the study period.

## Methods

### Study design

Consent was not obtained. Ethical approval for a non-consent, anonymised study was gained, REC reference 20/NI/0107, because strong safeguards against deductive disclosure of the identities of individuals with SARS-CoV-2 antibodies were inbuilt.

By initial design, we planned to analyse the results of 11,000 antenatal booking samples given at Imperial College Healthcare NHS Trust and Chelsea and Westminster NHS Foundation Trust from December 2019 to May 2020 to examine how the prevalence of seroreactivity to SARS-CoV-2 varied over time in this population. Samples were retrieved from the -20˚C storage freezer at North-West (NW) London Pathology, held in boxes of 100 samples in loose chronological order. Samples were thawed prior to vortexing each sample for 10–15 seconds. Where there was sufficient sample stored, 150µL of serum was aliquotted into a single well of a

96 well plate using the DS2 instrument (Dynex Technologies–USA). Once aliquotted, the plates were stored in a -80˚C freezer until they were transported on dry ice to the Molecular Diagnostics Unit (Imperial College London) for testing.

Prior to anonymization, samples were cross-classified by fortnight, age group (18–29 years, 30–34 years, 35–44 years), ethnicity (all white, all Asian, all Black, Other), and deprivation (most-deprived two IMD deciles 1–2, intermediate four deciles 3–6, least-deprived four deciles 7–10). During the aliquoting process, a small number of pairs of samples was identified to have come from the same person; the second of such samples to be aliquoted was marked as a duplicate and excluded from the formal analysis. We gave an undertaking not to report on any cross-classification for which the observed count was under 25. This undertaking was met by pooling across adjacent fortnights.

Testing of 5350 samples initially confirmed that sufficiently narrow confidence intervals for naïve prevalence estimates were obtained by testing around 500 samples per fortnight across the initial sampling period. Since samples prior to mid-February were reactive for anti-SARS-CoV-2 receptor binding domain (RBD) antibodies, a decision was taken in April 2021 to test around 500 samples per fortnight (rather than approximately all 1000 available samples held by NW London Pathology for the two north-west London trusts per fortnight) and focus the conserved test-resources on samples from fortnights earlier than late December 2019. The testing period was also extended to fortnights through September 2020 to allow comparison with national serosurveys running during this period. Additionally, due to the anticipated stabilised high prevalence in fortnights after May 2020, the number tested in those later fortnights was reduced to around 200.

In total, an aliquot of the stored sample was retrieved from 12,348 antenatal serum samples, collected over fortnightly intervals from fortnight 22 in 2019 (22 October– 4 November 2019) to fortnight 19 in 2020 (9–22 September 2020) with accompanying anonymised demographic information where available, as detailed above.

Following the initial rounds of testing, the continued presence of seroreactive samples across all tested fortnights in 2019 led us to seek additional funding to conduct further testing of booking samples from earlier in 2019. The earliest maternal booking samples available at the time were from June 2019; 1000 of these were aliquoted and tested. Further information on this validatory testing is provided below. As these data would be used only to aid interpretation of our main findings, it was not necessary (nor possible) to cross-classify samples by age group, ethnicity, and deprivation. Ethical approval for the original study was expanded to include the testing of these June 2019 samples, with the amendments to the study period classed as "non-substantial".

## Testing of samples

Samples were assigned a testing bin number, based on sample year, fortnight, age group, ethnicity group, and deprivation group (except for June 2019 samples which were only assigned a sample year and month). Testing was thereby conducted anonymously.

For the detection of total antibodies to SARS-CoV-2 receptor binding domain (anti-RBD) we used an in-house hybrid double antigen binding assay (DABA), the Imperial College London Hybrid DABA, which is a two-step sequential enzyme-linked immunosorbent assay (ELISA) and is accredited by the United Kingdom Accreditation Service (UKAS). The Imperial Hybrid DABA detects and quantifies total antibodies to SARS-CoV-2 anti-RBD [15]. This assay has a specificity greater than 99.6% (95% CI 99.6–100), defined by testing 825 serum samples that predated the COVID-19 pandemic, 100 of which were historic antenatal booking serum samples from 2018; and a sensitivity of 98.9% (95% CI 96.8–99.8) when evaluating 276

serum samples from individuals with RT-PCR-confirmed SARS-CoV-2 infection [16]. Further details are provided in Rosadas et al. [17].

### Defining seroreactivity

As set out in Rosadas et al. [17], the cutoff for seroreactivity was established by adding 0.1 to the average of optical density (OD) obtained for three negative controls assayed in each run. The signal-to-cutoff value, known as the binding ratio (BR), for each sample was determined by dividing the sample OD by the cutoff OD. A sample was generally considered seroreactive for SARS-CoV-2 anti-RBD when the BR was greater than or equal to 1. BRs between 0.8 and 1.2 were considered to display a weak signal (borderline non-seroreactive; borderline seroreactive).

### Statistical analysis

Data from October 2019 to September 2020 were cleaned to contain only "complete" observations in terms of BR, sampling fortnight, and categorisation into each of the age, ethnicity, and deprivation groups outlined previously.

Assuming that the number of reactive samples is a binomially-distributed random variable, prevalence of seroreactivity was estimated using the maximum likelihood estimate (MLE) as follows:

$$Prevalence\ \widehat{of\ sero}reactivity = \frac{Number\ of\ seroreactive\ observations}{Number\ of\ total\ observations}$$

The prevalence of seroreactivity was estimated per fortnight as well as for each age, ethnicity and deprivation groups over the entire sampling period, and is presented as the MLE with 95% exact binomial confidence interval. Prevalence of seroreactivity for the month of June 2019 was also estimated using this method.

Two-sided Fisher's exact tests and chi-squared tests were used to examine statistically significant changes in the proportions of seroreactive observations across fortnights.

Binomial logistic regression was used to quantify the relationship between seroreactivity and sampling fortnight, age, ethnicity and deprivation. The oldest age group, all-white ethnicity (ethnicity group with lowest prevalence), and most-deprived deprivation group were selected as the baseline groups. Per-protocol, the baseline sampling fortnight was the fortnight prior to lockdown. To test for interaction between ethnicity and deprivation, a further model with an interaction term between ethnicity and deprivation was considered, with the overall significance of this term determined by a chi-squared test.

Estimates of the incidence of seroreactivity per fortnight were derived from estimated prevalence of seroreactivity using a bootstrapping procedure. For each bootstrap sample, 11,256 bootstrap-observations in total were sampled with replacement from the available data (i.e. 11,256 actual observations), according to fortnightly groupings, and used to estimate prevalence of seroreactivity as detailed above. A shape-constrained P-spline was then fitted to the bootstrapped prevalence estimates to characterise the temporal trends in the data (S1 Fig in S1 File). The incidence per fortnight was simply the difference in the current and previous fortnight's modelled prevalence estimates. The incidence among susceptible persons per fortnight was then estimated by dividing the difference in the current and previous fortnight's modelled prevalence estimates by the proportion of population estimated to have been susceptible at the previous fortnight (1 minus the model estimate of the proportion seroreactive). Bootstrap sampling and analysis were repeated 1000 times. The reported central estimate is the median with

95% confidence interval constructed by taking the 2.5$^{th}$ and 97.5$^{th}$ quantiles of the bootstrap samples, respectively.

Estimated incidence of seroreactivity over the study period was compared to subsequent seven-day average daily hospital admission and death rates in London [1], akin to Riley et al. [18]. Hospital admissions and deaths were shifted by a lag of $L_H$ and $L_D$ days, respectively, to capture the delay between infection, the development of seroreactivity, hospitalisation, and death, and scaled by a value of $\beta_H$ and $\beta_D$ to capture the different scales of these time series. $L_H$ and $L_D$ were allowed to take integer values from 0 to 40 (days). For each integer value from 0 to 40 days, individual linear regression models estimated incidence per fortnight on the appropriately-lagged daily hospital admissions and death rates, weighted proportional to the inverse of the variance of the incidence estimates. The best-fitting lag was selected according to the model which minimised the weighted sum of squared errors (SSE). Conditional on the best-fitting lag, the scaling parameter was the slope estimated in the regression model.

### Validation

To validate our testing by DABA in samples from June 2019, all seroreactive samples were re-tested blind with 2 consecutive, negative controls for each positive sample, in the United Kingdom Health Security Agency (UKHSA) Colindale laboratory using their nucleotide protein (NP) and spike protein S1 subunit (S1) capture assays, with and without blocking for seasonal coronaviruses.

## Results

Fifty-five duplicates were removed from the 12,348 retrieved samples. Of the 12,293 non-duplicated retrieved samples from October 2019 to September 2020, 11,256 (92%) had binding ratio value, sampling fortnight and were categorised into each of the relevant age, ethnicity and IMD groups and so were included in the analysis.

Seven hundred of the 11,256 samples were seroreactive (defined by a BR ≥ 1), resulting in an overall prevalence of seroreactivity estimate of 6.2% (95% CI 5.8% - 6.7%). The vast majority of seroreactive samples (665; 95%) were categorised as 'clearly seroreactive' (BR ≥ 1.2) (S1 Table in S1 File).

The median BRs among seroreactive observations per fortnight remained at low levels across October 2019 –February 2020 (Fig 1). From late February 2020, there is a sharp increase in the median BRs per fortnight through late April 2020. Thereafter there is a decrease, although BR values do not fall to the level observed at the beginning of the sampling period. Among non-seroreactive observations, median BRs remained fairly constant across fortnights throughout the main sampling period (Fig 1).

### Prevalence by demographic variables and by fortnight

There was heterogeneity in the sample size across ethnicity and IMD groups, even more than anticipated, with fewest observations seen for the youngest age-group (18–29 years: 25.9%), black ethnicity (7.4%) and the most deprived IMD-group (IMD 1–2: 13.7%). Overall, the prevalence of seroreactivity decreased with age; was more than twice as common in pregnant black women compared to pregnant white women; and increased as deprivation increased (Table 1).

Seroreactive samples were found in the earliest fortnight of the primary study period (October 2019 –September 2020): fortnight 22 of 2019 (22 October– 4 November 2019) (Fig 2; Table 2). Each age, ethnicity (except the all-black group) and deprivation group was represented among the earliest seroreactive samples.

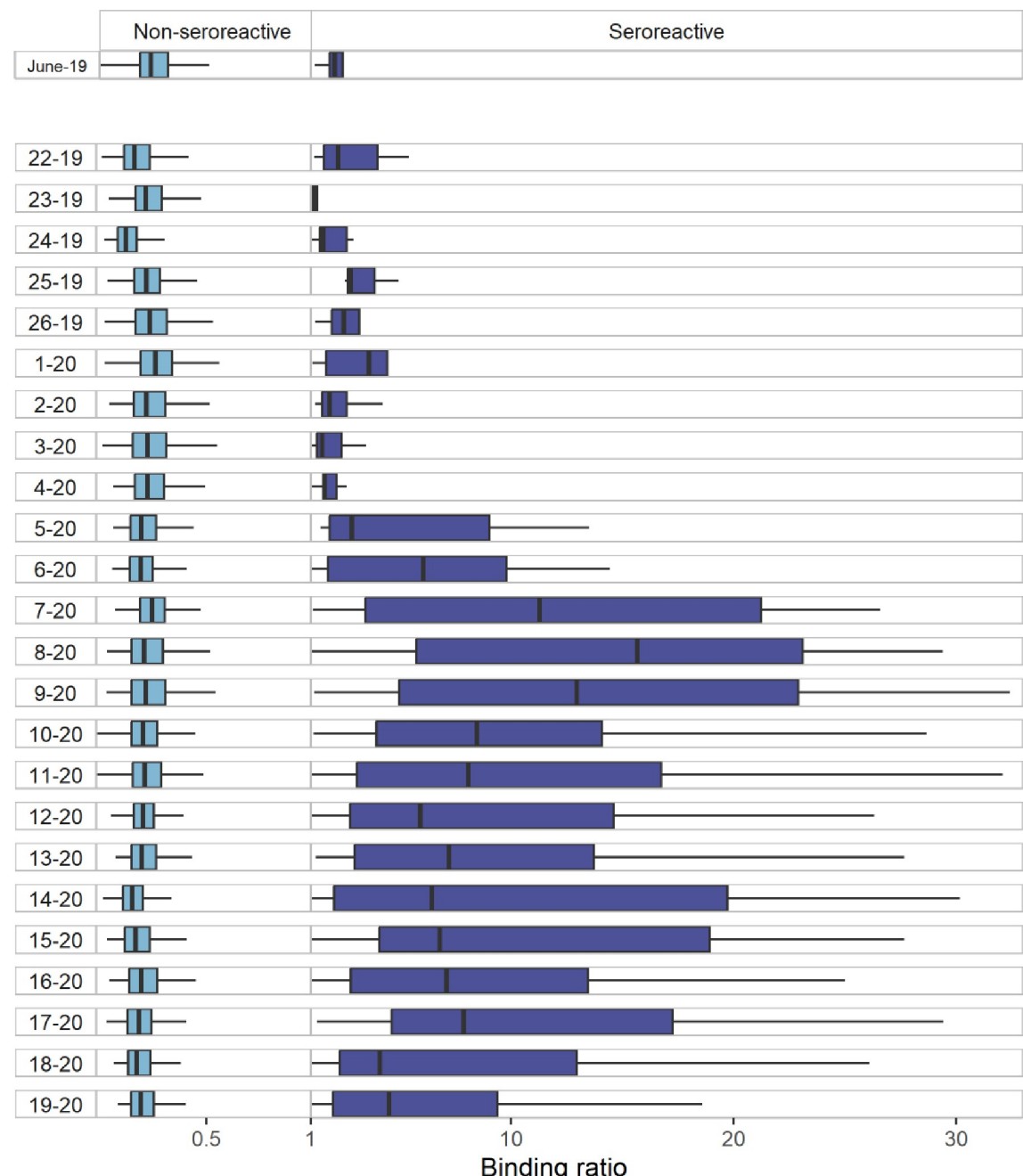

**Fig 1.** Binding ratio values among non-seroreactive (left) and seroreactive (right) observations per fortnight (October 2019-September 2020) and per month for June 2019. Note that the assay approximately 22–23 non-linear. From left to right, lines on the boxplot indicate: 1.5 times the interquartile range (IQR) less than the first quartile, first quartile, first quartile, median third quartile and 1.5 times the IQR greater than the third quartile. Outliers, defined as any point outwith the lower and upper bounds described have been removed so as not to obscure or distort the presentation of the other results.

Estimated prevalence of seroreactivity increased substantially across February–May 2020 (Fig 2; Table 2) coincident with an increase in national confirmed case numbers, hospitalisations and deaths attributable to COVID-19 [1]. Despite reasonably large sample sizes, prevalence of seroreactivity estimates in 2019 were low and somewhat volatile over time, which four-weekly summaries counteracted (Table 2).

**Table 1. Demographics of the study sample.** The number of total observations and the number of seroreactive observations per group within each demography variable across the primary study period (October 2019 –September 2020).

| Demography variable | Group | Total observations (n) | Seroreactive observations (x) | Estimated prevalence of seroreactivity ($\hat{\hat{s}}$) (MLE (95% exact binomial confidence intervals)) |
|---|---|---|---|---|
| Age | 18–29 | 2930 | 226 | 7.7% (6.8% - 8.7%) |
| | 30–34 | 4135 | 256 | 6.2% (5.5% - 7.0%) |
| | 35–44 | 4226 | 218 | 5.2% (4.5% - 5.9%) |
| Ethnicity | All-black | 839 | 87 | 10.4% (8.4% - 12.6%) |
| | All-Asian | 2016 | 160 | 8.0% (6.8% - 9.2%) |
| | All-white | 4613 | 208 | 4.5% (3.9% - 5.2%) |
| | Other | 3823 | 245 | 6.4% (5.7% - 7.2%) |
| Deprivation | IMD deciles 1–2 | 1546 | 120 | 7.8% (6.5% - 9.2%) |
| | IMD deciles 3–6 | 6492 | 427 | 6.6% (6.0% - 7.2%) |
| | IMD deciles 7–10 | 3253 | 153 | 4.7% (4.0% - 5.5%) |

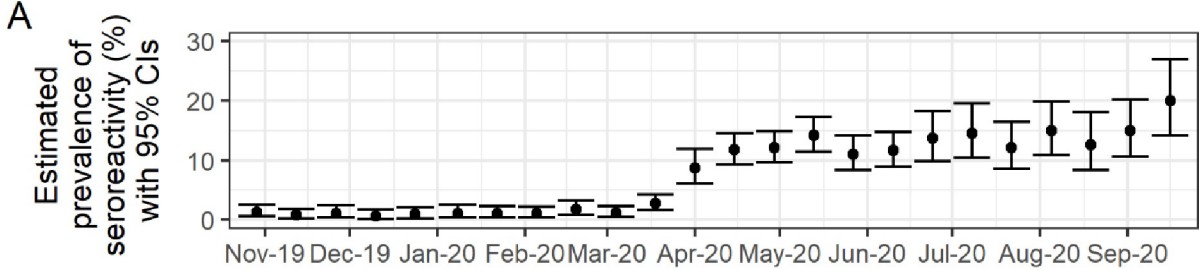

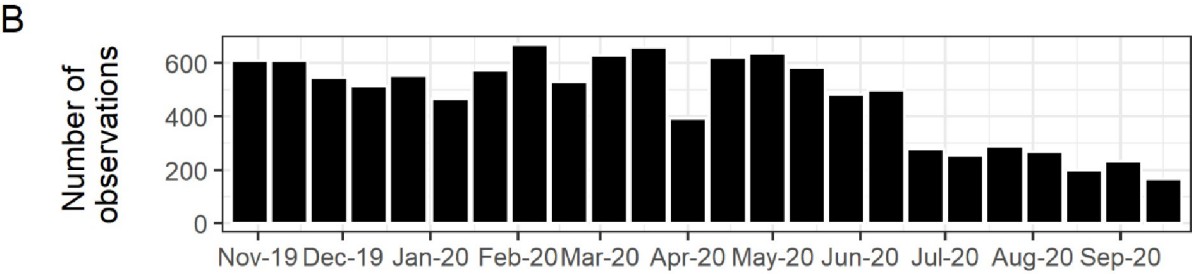

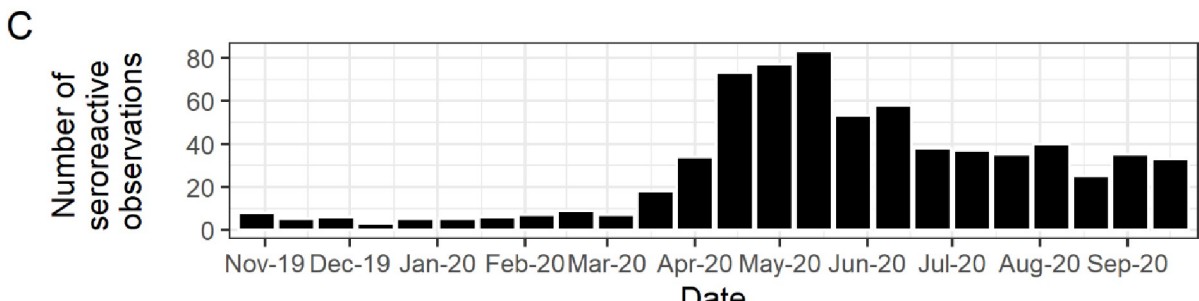

**Fig 2. Prevalence of seroreactivity over time.** (A) Estimated prevalence of seroreactivity over time with exact binomial 95% confidence intervals. (B) Total number of observations per fortnight. (C) Number of seoreactive observations per fortnight.

**Table 2. Sample size and naïve prevalence of seroreactivity over time.** The total number of observations and the number of seroreactive observations per fortnight and per four-week period using a seroreactive threshold of BR value $\geq 1$ in the primary study period (and per month in June 2019). This is used to calculate naïve estimates of the prevalence of seroreactivity with exact binomial 95% confidence intervals. Note fortnight 26 of 2019 spans across 15 rather than 14 days.

| Collection fortnight (fortnight-year) | Calendar date | Fortnightly | | | 4-week period | | |
|---|---|---|---|---|---|---|---|
| | | Total observations ($n$) | Seroreactive observations ($x$) | Estimated prevalence of seroreactivity $(\hat{s})$ (MLE (95% exact binomial confidence intervals)) | Total observations ($n$) | Seroreactive observations ($x$) | Estimated prevalence of seroreactivity $(\hat{s})$ (MLE (95% exact binomial confidence intervals)) |
| June-2019 | 01/06/2019–30/06/2019 | | | | 1000 | 8 | 0.8% (0.3% - 1.6%) |
| 22–2019 | 22/10/2019–04/11/2019 | 608 | 8 | 1.3% (0.6% - 2.6%) | 1218 | 13 | 1.1% (0.6% - 1.8%) |
| 23–2019 | 05/11/2019–18/11/2019 | 610 | 5 | 0.8% (0.3% - 1.9%) | | | |
| 24–2019 | 19/11/2019–02/12/2019 | 546 | 6 | 1.1% (0.4% - 2.4%) | 1060 | 9 | 0.8% (0.4% - 1.6%) |
| 25–2019 | 03/12/2019–16/12/2019 | 514 | 3 | 0.6% (0.1% - 1.7%) | | | |
| 26–2019 | 17/12/2019–31/12/2019 | 554 | 5 | 0.9% (0.3% - 2.1%) | 1021 | 10 | 1.0% (0.5% - 1.8%) |
| 1–2020 | 01/01/2020–14/01/2020 | 467 | 5 | 1.1% (0.4% - 2.5%) | | | |
| 2–2020 | 15/01/2020–28/01/2020 | 572 | 6 | 1.1% (0.4% - 2.3%) | 1241 | 13 | 1.0% (0.6% - 1.8%) |
| 3–2020 | 29/01/2020–11/02/2020 | 669 | 7 | 1.1% (0.4% - 2.1%) | | | |
| 4–2020 | 12/02/2020–25/02/2020 | 530 | 9 | 1.7% (0.8% - 3.2%) | 1158 | 16 | 1.4% (0.8% - 2.2%) |
| 5–2020 | 26/02/2020–10/03/2020 | 628 | 7 | 1.1% (0.5% - 2.3%) | | | |
| 6–2020 | 11/03/2020–24/03/2020 | 659 | 18 | 2.7% (1.6% - 4.3%) | 1051 | 52 | 4.9% (3.7% - 6.4%) |
| 7–2020 | 25/03/2020–07/04/2020 | 392 | 34 | 8.7% (6.1% - 11.9%) | | | |
| 8–2020 | 08/04/2020–21/04/2020 | 622 | 73 | 11.7% (9.3% - 14.5%) | 1259 | 150 | 11.9% (10.2% - 13.8%) |
| 9–2020 | 22/04/2020–05/05/2020 | 637 | 77 | 12.1% (9.7% - 14.9%) | | | |
| 10–2020 | 06/05/2020–19/05/2020 | 584 | 83 | 14.2% (11.5% - 17.3%) | 1066 | 136 | 12.8% (10.8% - 14.9%) |
| 11–2020 | 20/05/2020–02/06/2020 | 482 | 53 | 11.0% (8.4% - 14.1%) | | | |
| 12–2020 | 03/06/2020–16/06/2020 | 499 | 58 | 11.6% (9.0% - 14.8%) | 777 | 96 | 12.4% (10.1% - 14.9%) |
| 13–2020 | 17/06/2020–30/06/2020 | 278 | 38 | 13.7% (9.9% - 18.3%) | | | |
| 14–2020 | 01/07/2020–14/07/2020 | 254 | 37 | 14.6% (10.5% - 19.5%) | 542 | 72 | 13.3% (10.5% - 16.4%) |
| 15–2020 | 15/07/2020–28/07/2020 | 288 | 35 | 12.2% (8.6% - 16.5%) | | | |
| 16–2020 | 29/07/2020–11/08/2020 | 267 | 40 | 15.0% (10.9% - 19.8%) | 465 | 65 | 14.0% (11.0% - 17.5%) |
| 17–2020 | 12/08/2020–25/08/2020 | 198 | 25 | 12.6% (8.3% - 18.1%) | | | |
| 18–2020 | 26/08/2020–08/09/2020 | 233 | 35 | 15.0% (10.7% - 20.3%) | 398 | 68 | 17.1% (13.5% - 21.2%) |
| 19–2020 | 09/09/2020–22/09/2020 | 165 | 33 | 20.0% (14.2% - 26.9%) | | | |

While there was an overall increase in the prevalence of seroreactivity across June–September 2020, some of the fortnightly heterogeneity in prevalence estimates in this period was reflective of the comparatively smaller sample size (Fig 2; Table 2). This was also overcome using four-weekly summaries. For example, prevalence of seroreactivity significantly increased from 12.4% (95% CI 10.1% to 14.9%) in June 2020 (fortnights 12–13) to 17.1% (95% CI 13.5% to 21.2%) in September 2020 (fortnights 18–19) (chi-squared test: $X^2 = 4.52$; n = 1175; p = 0.034).

Eight out of 1000 samples from the additional June 2019 samples were also found to be seroreactive. The median BR of these eight samples was similar to those observed across the initial fortnights of the primary study period (Fig 1). Two-sided Fisher's exact tests indicated no statistically significant changes in the odds of observing a seroreactive sample across each fortnight between fortnight 22 of 2019 and fortnight 5 of 2020 in comparison to June 2019, suggesting no evidence of community transmission of SARS-CoV-2 within our sample until after January 2020 (S1 File).

**Multifactorial logistic regression.** Women in the youngest age category were significantly more likely to be seroreactive than women in the oldest age category (p = 0.004). Minority ethnicity groups were associated with significantly increased seroreactivity compared to the all-white group (p<0.001 for all-black, all-Asian and other groups each compared to all-white) (Table 3). Relative deprivation was associated with seroreactivity, but significantly so only for the most deprived IMD sub-grouping in comparison to the least deprived (p = 0.020).

The 10 fortnights between 22 October 2019 and 10 March 2020 (which precede the reference fortnight of 11 to 24 March) had significantly reduced levels of seroreactivity. On the other hand, all fortnights after the reference fortnight (13 fortnights between 25 March 2020 and 22 September 2020) had significantly increased levels of seroreactivity, compared to the reference fortnight.

The interaction term between ethnicity and deprivation groups was not statistically significant (chi-squared test: $X^2 = 4.80$; degrees of freedom = 6; overall p = 0.570).

## Incidence of seroreactivity over time

The estimated incidence of seroreactivity per fortnight remained around zero until February 2020 (Fig 3). After this time, there was a notable increase in estimated incidence per fortnight. This reached a peak between 11 March and 7 April 2020, corresponding to fortnights 6 and 7 of 2020, which encompasses the implementation of the first national lockdown on 23 March 2020. Incidence was then estimated to fall until around mid-August 2020 after which point it began to increase again. The shape of the incidence curve derived from estimated prevalence of seroreactivity reflects the trends displayed in daily hospital admission and death rates of COVID-19-confirmed cases in London 10 and 13 days later, respectively (Fig 3). Estimates of the incidence among susceptible persons is presented in the S1 File.

## Validation of seroreactivity

Of the 8 samples from June 2019 which were seroreactive on DABA, none were found to be reactive on UKHSA NP or S1 capture assays. Of the 16 non-seroreactive samples on DABA (corresponding negative controls from June 2019), 2 were seroreactive on NP Immunoglobulin G (IgG) assay but were non-seroreactive after blocking for seasonal coronaviruses.

Internal validation found 1 of the 8 samples from June 2019 which were seroreactive on DABA was positive on Immunoglobulin M (IgM) S1 capture assay (S3 Table in S1 File).

**Table 3. Results of logistic regression for seroreactivity using demographic variables.** Coefficient estimates, standard errors and p-values for the logistic regression model with age group, ethnicity, IMD and fortnight of sample (across the primary study period October 2019 –September 2020) as predictors of seroreactivity. The dotted line indicates the timing of the reference fortnight.

| Variable | Interpretation | Estimate | Standard Error | Adjusted Odds Ratio | p-value |
|---|---|---|---|---|---|
| (Intercept) | 35–44 | -3.93 | 0.27 | | |
| | All-white | | | | |
| | IMD decile group 1–2 | | | | |
| | 11/03/2020–24/03/2020 | | | | |
| Age | 18–29 | 0.30 | 0.10 | 1.35 | 0.004 |
| | 30–34 | 0.12 | 0.10 | 1.13 | 0.223 |
| Ethnicity | All-black | 0.91 | 0.14 | 2.47 | <0.001 |
| | All-Asian | 0.57 | 0.11 | 1.77 | <0.001 |
| | Other | 0.40 | 0.10 | 1.49 | <0.001 |
| Deprivation | IMD decile 3–6 | -0.08 | 0.11 | 0.92 | 0.461 |
| | IMD decile 7–10 | -0.32 | 0.14 | 0.73 | 0.020 |
| Fortnight | 22/10/2019–04/11/2019 | -0.74 | 0.43 | 0.48 | 0.084 |
| | 05/11/2019–18/11/2019 | -1.23 | 0.51 | 0.29 | 0.016 |
| | 19/11/2019–02/12/2019 | -0.92 | 0.48 | 0.40 | 0.054 |
| | 03/12/2019–16/12/2019 | -1.61 | 0.63 | 0.20 | 0.010 |
| | 17/12/2019–31/12/2019 | -1.17 | 0.51 | 0.31 | 0.022 |
| | 01/01/2020–14/01/2020 | -0.95 | 0.51 | 0.39 | 0.061 |
| | 15/01/2020–28/01/2020 | -0.97 | 0.48 | 0.38 | 0.041 |
| | 29/01/2020–11/02/2020 | -0.98 | 0.45 | 0.38 | 0.030 |
| | 12/02/2020–25/02/2020 | -0.43 | 0.41 | 0.65 | 0.295 |
| | 26/02/2020–10/03/2020 | -0.89 | 0.45 | 0.41 | 0.048 |
| | 25/03/2020–07/04/2020 | 1.24 | 0.30 | 3.47 | <0.001 |
| | 08/04/2020–21/04/2020 | 1.57 | 0.27 | 4.79 | <0.001 |
| | 22/04/2020–05/05/2020 | 1.59 | 0.27 | 4.88 | <0.001 |
| | 06/05/2020–19/05/2020 | 1.78 | 0.27 | 5.93 | <0.001 |
| | 20/05/2020–02/06/2020 | 1.52 | 0.28 | 4.58 | <0.001 |
| | 03/06/2020–16/06/2020 | 1.55 | 0.28 | 4.73 | <0.001 |
| | 17/06/2020–30/06/2020 | 1.75 | 0.30 | 5.78 | <0.001 |
| | 01/07/2020–14/07/2020 | 1.84 | 0.30 | 6.32 | <0.001 |
| | 15/07/2020–28/07/2020 | 1.62 | 0.30 | 5.07 | <0.001 |
| | 29/07/2020–11/08/2020 | 1.87 | 0.30 | 6.47 | <0.001 |
| | 12/08/2020–25/08/2020 | 1.67 | 0.32 | 5.31 | <0.001 |
| | 26/08/2020–08/09/2020 | 1.85 | 0.30 | 6.39 | <0.001 |
| | 09/09/2020–22/09/2020 | 2.26 | 0.31 | 9.56 | <0.001 |

## Discussion

Our study presents a novel analysis of historic antenatal serum samples from an ethnically and socially diverse population in north-west London from late October 2019 to September 2020 in London, during which time SARS-CoV-2 was first detected. There were three direct flights per week from Wuhan to London Heathrow airport, in north-west London, until air travel from China to the UK was suspended on 29 January 2020 [19].

We found reactive samples throughout our testing period, but no evidence of sustained community transmission until after January 2020, assuming a seroconversion period of approximately 2–3 weeks. The temporal trends displayed in our estimated incidence of seroreactivity prevalence matched those observed in daily hospital admission and death rates in London that followed 10 and 13 days later, respectively [1].

A

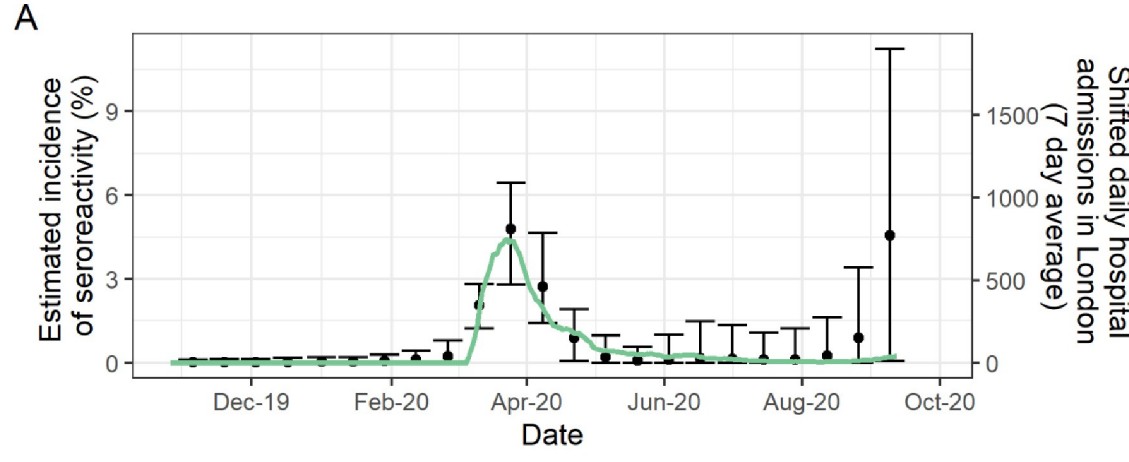

B

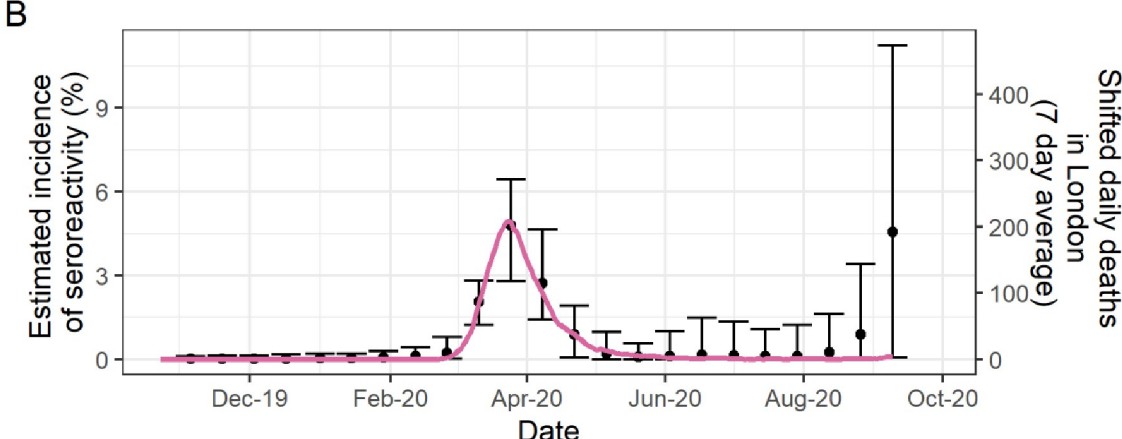

**Fig 3. Estimated incidence of seroreactivity over time with 95% bootstrapped confidence intervals.** (A) Seven-day averages of daily hospital admissions in London' shifted by 10 days and scaled by 0.006. (B) Seven-day averages of daily deaths in London' shifted by 13 days and scaled by 0.024. See Methods: Statistical analysis for further details.

Our results showed higher prevalence of seroreactivity to SARS-CoV-2 in non-white ethnicity groups, in the most deprived group and in younger age groups, in accordance with findings of other research groups [20–22]. The absence of significant interaction between ethnicity group and IMD deprivation group indicated no evidence in the data that the ethnicity-risk was exacerbated by relative deprivation. [23] Ethnicity itself was a key factor in SARS-CoV-2 infection rates in our study population.

Since the assay used for this study has a high specificity of 100% (95% CI 99.6–100) and sensitivity of 98.9% (95% CI 96.8–99.8) [15] our estimates for the prevalence of seroreactivity were not adjusted for sensitivity and specificity due to the negligible impact that this had on the results.

Our results did not identify the introduction into the United Kingdom of the highly transmissible SARS-CoV-2 variant, first detected in Wuhan. The external testing of our June 2019 seroreactive samples from 2019 indicate that the seroreactivity in the early samples was likely attributable to cross-reactivity with antibody to related viruses in these antenatal sera, as described in other studies [24, 25]. Incidence estimates depend on changes in changes in prevalence and so are unaffected by a low background level of cross-reactivity.

The end of our study period briefly coincided (for four fortnights) with the beginning of the England-wide Real-time Assessment of Community Transmission (REACT-2) study

which had an initial response rate of 34% [22]. When comparing our estimates to a subset of the REACT-2 adult respondents matching our demographic variables, our estimates were consistent with the proportion antibody positive in REACT-2 (among female participants from London and aged 18 to 44 years) in their in June–July 2020 surveillance but not thereafter when the REACT-2 proportions declined (S3 Fig in S1 File). However, REACT-2 used a different assay. Antibody responses wane with elapsed time since infection, but may do so differently between assays [22]. Adjusting for sensitivity and specificity of the different assay, using the method by Rogan et al. [26], did not explain the observed differences. Nonetheless, our estimates clearly captured the increase in seroreactivity prevalence in the first half of 2020, as reflected also in the first REACT-2 survey (June–July 2020).

Other serosurveillance studies have not provided robust insights into community transmission in late 2019 and early 2020. Lumley et al. [11] reported on 1,000 antenatal samples from April and May 2020, of which 53 were reactive. Testing of samples from blood donors across the UK has also been undertaken. Thompson et al. [9] tested 3,500 blood donor samples in Scotland across March to May 2020, of which 111 were reactive but none of the 500 obtained on 17 March 2020; and Public Health England tested 1000 blood donor samples per region per week, beginning from late March 2020. [10] Additionally, Dickson et al. [8] analysed 4751 residual blood samples from Scottish regional biochemistry laboratories estimating a seroprevalence of 4.3% across April–June 2020. We note that the population of blood donors is typically less socially and ethnically diverse than pregnant women, two risk factors which have been shown to be associated with risk of infection of SARS-CoV-2.

Other published historical seroprevalence studies for detection of SARS-CoV-2 have often focussed on the timing of introduction rather than using their data to derive estimates of incidence. An historical seroprevalence study tested stored samples held on 959 patients in a lung cancer treatment trial from sites across Italy, detecting anti-RBD IgG and IgM antibodies to SARS-CoV-2 in September 2019 in 14% of samples [27]. Surprisingly, seroprevalence in this cohort fell to 2.8% in January 2020, rising to 20% in February 2020. The authors did not describe assay performance and background detection rates. Furthermore, their results were unable to be confirmed in an independent laboratory, at the request of the World Health Organization, thus reducing confidence in their observations [28].

The COVID-19 pandemic has underlined the importance of robust surveillance systems to detect community transmission as early as possible. Alternative, less-traditional means by which to detect public health impact have been explored throughout various epidemics. For example, during the COVID-19 pandemic all-cause excess mortality has been used to examine the possibility of wider transmission than those captured in official statistics as well as deaths caused indirectly by COVID-19's impacts (for example, through overwhelmed healthcare systems). One such study in England and Wales showed an increase in excess deaths from March 2020, which matches our finding that widespread community transmission was not observed until after January 2020 [29].

Antenatal booking samples from at least two years ago (March 2020, at time of writing) are still currently stored throughout the UK (and potentially overseas [30]) and could allow mapping of the introduction and spread of the virus in this population over time with demographic characterisation. However, storage of these samples is not routinely required beyond 2 years. 'Pregnant women's increased susceptibility for some respiratory viruses make them a crucial group for surveillance, however the evidence for increased susceptibility to SARS-CoV-2 in pregnancy is mixed [31]. Advice for pregnant women to shield may affect the generalisability of our data to other populations, however these samples were taken at around 10–12 weeks gestation when the majority of women would have only recently have become aware of pregnancy.'

In future epidemics, stored maternal booking serum samples from areas centred on UK travel hubs could be used, once reliable assays have been developed, to give retrospective estimates of early incidence rates; to identify populations at risk during the earliest stages of transmission and hence to track the emergence of a new virus. Such estimates would complement any blood donor serosurveillance undertaken. Surveillance for historical SARS-CoV-2 seroreactivity in blood donors could, even now, usefully compare 5,000 to 10,000 stored samples from each of June and December in 2018 versus 2019 to allow for donors' lower-risk while robustly investigating seasonality and cross-reactivity.

The data we report show the value of historical maternal seroprevalence studies in exploring the dynamics of the SARS-CoV-2 pandemic and pre-pandemic periods. The maternal serum samples from north-west London from October 2019 onwards remain in storage and could, in combination with sera stored in other regions of the UK and around the world, offer fascinating insights into the spread of the SARS-CoV-2 virus in different regional populations. As an anonymised tool for the future, these samples are a comprehensive rolling serum bank which stands ready to be activated to understand new and emerging viral threats. This could be an important component of ongoing surveillance at a time when some SARS-CoV-2 infection-surveillance programmes are being wound down [32, 33].

## Supporting information

**S1 File.**
(DOCX)

## Acknowledgments

We wish to thank several individuals who helped with this study. Infrastructure support was provided by the NIHR Imperial College Biomedical Research Centre, Ben Glampson at Imperial College Healthcare Trust (ICHT) and Stephen Cole at Chelsea and Westminster Healthcare Foundation Trust (CWHFT). Tina Cotzias and Natasha Mohammed for acted as PIs for ChelWest Foundation NHS Trust. Jonathan Dennis supported with initial literature searches and prepared the study for ethical review. Sarah Littell at NW London Pathology identified and aliquoted samples. Victoria Male provided helpful discussions about immune responses in pregnancy. We thank the REal-time Assessment of Community Transmission (REACT) Study investigators for sharing antibody prevalence data. We are grateful to Samreen Ijaz at UKHSA for conducting validation testing on 2019 seroreactive samples. We are grateful to Wendy Barclay, Peter Horby and Peter Openshaw for their comments on this manuscript.

## Author Contributions

**Conceptualization:** Edward Mullins.

**Formal analysis:** Ruth McCabe, Sheila M. Bird, Christl A. Donnelly.

**Funding acquisition:** Edward Mullins, Sheila M. Bird, Paul Randell, Lesley Regan, Christl A. Donnelly.

**Investigation:** Edward Mullins, Ruth McCabe, Sheila M. Bird, Paul Randell, Marcus J. Pond, Eleanor Parker, Myra McClure, Christl A. Donnelly.

**Methodology:** Edward Mullins, Paul Randell, Lesley Regan, Myra McClure, Christl A. Donnelly.

**Project administration:** Edward Mullins, Marcus J. Pond, Eleanor Parker.

**Resources:** Paul Randell, Marcus J. Pond, Eleanor Parker.

**Writing – original draft:** Edward Mullins, Ruth McCabe.

**Writing – review & editing:** Edward Mullins, Ruth McCabe, Sheila M. Bird, Paul Randell, Marcus J. Pond, Lesley Regan, Eleanor Parker, Myra McClure, Christl A. Donnelly.

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
