## [Decision Letter · Decision Letter 0]

12 Aug 2022

PONE-D-22-11657Tracking the incidence and risk factors for SARS-CoV-2 infection using historical maternal booking serum samplesPLOS ONE

Dear Author,

Thank you for submitting your manuscript to PLOS ONE. After careful consideration, we feel that it has merit but does not fully meet PLOS ONE’s publication criteria as it currently stands. Therefore, we invite you to submit a revised version of the manuscript that addresses the points raised during the review process.

We look forward to receiving your revised manuscript.

Kind regards,

Atalay Goshu Muluneh, MPH

Academic Editor

PLOS ONE

Journal Requirements:

This research was partly funded by Community Jameel and the Imperial President’s Excellence Fund. https://www.imperial.ac.uk/jameel-institute/ 

This research was partly funded by Community Jameel and the Imperial President’s Excellence Fund.

EM was funded by an NIHR Academic Clinical Lecturer post until February 2021. 

RM and CAD acknowledge the NIHR HPRU in Emerging and Zoonotic Infections, a partnership between Public Health England (PHE), University of Oxford, University of Liverpool and Liverpool School of Tropical Medicine [grant number NIHR200907]. CAD also acknowledges the MRC Centre for Global Infectious Disease Analysis [grant number MR/R015600/1], which is jointly funded by the UK Medical Research Council (MRC) and the UK Foreign, Commonwealth & Development Office (FCDO), under the MRC/FCDO Concordat agreement and is also part of the EDCTP2 programme supported by the European Union (EU). 

EP, MM and RST acknowledge support from NIHR CV220-111: Serological detection of past SARS-CoV-2 infection by non-invasive sampling for field epidemiology and quantitative antibody detection and from departmental funds.

Disclaimer: “The views expressed are those of the authors and not necessarily those of the United Kingdom (UK) Department of Health and Social Care, EU, FCDO, MRC, National Health Service, NIHR, or PHE. The funding bodies had no role in the design of the study, analysis and interpretation of data and in writing the manuscript.”

However, funding information should not appear in the Acknowledgments section or other areas of your manuscript. We will only publish funding information present in the Funding Statement section of the online submission form. 

This research was partly funded by Community Jameel and the Imperial President’s Excellence Fund. https://www.imperial.ac.uk/jameel-institute/ 

EM was funded by an NIHR Academic Clinical Lecturer post until February 2021. https://www.nihr.ac.uk/explore-nihr/academy-programmes/integrated-academic-training.htm 

RM and CAD acknowledge the NIHR HPRU in Emerging and Zoonotic Infections, a partnership between Public Health England (PHE), University of Oxford, University of Liverpool and Liverpool School of Tropical Medicine [grant number NIHR200907]. 

CAD also acknowledges the MRC Centre for Global Infectious Disease Analysis [grant number MR/R015600/1], which is jointly funded by the UK Medical Research Council (MRC) and the UK Foreign, Commonwealth & Development Office (FCDO), under the MRC/FCDO Concordat agreement and is also part of the EDCTP2 programme supported by the European Union (EU). 

EP, MM and RST acknowledge support from NIHR CV220-111: Serological detection of past SARS-CoV-2 infection by non-invasive sampling for field epidemiology and quantitative antibody detection and from departmental funds.

Infrastructure support was provided by the NIHR Imperial College Biomedical Research Centre.

EM is an academic editor at PLOS One.

MM is listed as an inventor in IPR filings for the Imperial Hybrid DABA used in this analysis. Please see United Kingdom Patent Application No. 2011047.4 for “SARS-CoV-2 antibody detection assay”.

SMB is member of both Royal Statistical Society’s COVID-19 Taskforce and Working Group on Diagnostic Tests. SMB serves on UKHSA/DHSC’s Testing Initiatives Evaluation Board (January 2021 to present). All other authors declare no Conflicts of Interest. 

Additional Editor Comments:

Title: Tracking the incidence and risk factors for SARS-CoV-2 infection using historical maternal booking serum samples

Editor comment:

1. Please abide by the journal guideline

2. Check the grammar

3. Data availability is mandatory so please attach the data

Reviewer 1

In this manuscript, Mullins and McCabe et al. have estimated the prevalence of seroreactivity of total antibodies to the severe acute respiratory syndrome coronavirus 2 (SARS-CoV-2) receptor-binding domain (anti-RBD) in stored antenatal serum samples over time between COVID-19 pre-pandemic and pandemic periods. The authors reported a higher prevalence of seroreactivity to SARS-CoV-2 on Sept 2019 compared to mid-Feb 2020. Overall it is an interesting study representing a retrospective analysis of the prevalence of seroreactivity over time and its association with variables. My comments to the authors are:

1. In the method, in the logistic regression, have you controlled potential confounders (such as BMI, previous history of flu infections, flu vaccination status etc.) potentially affected the outcome variable in the analysis?

2. In the method ‘Defining seroreactivity’ section, please provide rationale and/or reference why the binding ratio (BR) ≥ 1 was considered as seroreactive for SARS-CoV-2 anti-RBD.

3. In the result, table 3, is it adjusted or unadjusted odds?

4. Some minor errors in the use of English throughout, i.e, missing/incorrect use of articles and prepositions - further proofreading required.

Reviewer 2

First of all, I would like to thank all authors.... this manuscript titled “Tracking the incidence and risk factors for SARS-CoV-2 infection using historical maternal booking serum samples" is easy to follow and well written. However, I have only one comment:

Could you please follow PLOS ONE guidelines, such as line number and reference style?

Reviewer 3 comment

1. Would like to hear more about the choice of sample size prior to starting the study. Maybe mention the possible limitations of generalizability of maternal cohorts when pregnant women may be more susceptible to infection.

Reviewers' comments:

Reviewer's Responses to Questions

**Comments to the Author**

1. Is the manuscript technically sound, and do the data support the conclusions?

Reviewer #1: Yes

Reviewer #2: Yes

Reviewer #3: Yes

2. Has the statistical analysis been performed appropriately and rigorously? 

Reviewer #1: Yes

Reviewer #2: Yes

Reviewer #3: I Don't Know

3. Have the authors made all data underlying the findings in their manuscript fully available?

Reviewer #1: Yes

Reviewer #2: Yes

Reviewer #3: Yes

4. Is the manuscript presented in an intelligible fashion and written in standard English?

Reviewer #1: Yes

Reviewer #2: Yes

Reviewer #3: Yes

5. Review Comments to the Author

Reviewer #1: In this manuscript, Mullins and McCabe et al. have estimated the prevalence of seroreactivity of total antibodies to the severe acute respiratory syndrome coronavirus 2 (SARS-CoV-2) receptor-binding domain (anti-RBD) in stored antenatal serum samples over time between COVID-19 pre-pandemic and pandemic periods. The authors reported a higher prevalence of seroreactivity to SARS-CoV-2 on Sept 2019 compared to mid-Feb 2020. Overall it is an interesting study representing a retrospective analysis of the prevalence of seroreactivity over time and its association with variables. My comments to the authors are:

1. In the method, in the logistic regression, have you controlled potential confounders (such as BMI, previous history of flu infections, flu vaccination status etc.) potentially affected the outcome variable in the analysis?

2. In the method ‘Defining seroreactivity’ section, please provide rationale and/or reference why the binding ratio (BR) ≥ 1 was considered as seroreactive for SARS-CoV-2 anti-RBD.

3. In the result, table 3, is it adjusted or unadjusted odds?

4. Some minor errors in the use of English throughout, i.e, missing/incorrect use of articles and prepositions - further proofreading required.

Reviewer #2: First of all, I would like to thank all authors.... this manuscript titled “Tracking the incidence and risk factors for SARS-CoV-2 infection using historical maternal booking serum samples" is easy to follow and well written. However, I have only one comment:

Could you please follow PLOS ONE guidelines, such as line number and reference style?

Reviewer #3: Would like to hear more about the choice of sample size prior to starting the study. Maybe mention the possible limitations of generalizability of maternal cohorts when pregnant women may be more susceptible to infection.

6. PLOS authors have the option to publish the peer review history of their article (what does this mean?). If published, this will include your full peer review and any attached files.

Reviewer #1: **Yes: **Md Jahangir Alam

Reviewer #2: No

Reviewer #3: No

---

## [Author Response · Author response to Decision Letter 0]

17 Aug 2022

Please see cover letter. 

Editor comments:

1. Please abide by the journal guideline

Format adjusted in-line with journal guidance. 

2. Check the grammar 

Thank you, I have re-read and I can’t see any further errors. I may be insensitive to my own mistakes. 

3. Data availability is mandatory so please attach the data

As per the manuscript, study data is available open access at https://github.com/ruthmccabe/historic_antenatal_serostudy

Reviewer 1

In this manuscript, Mullins and McCabe et al. have estimated the prevalence of seroreactivity of total antibodies to the severe acute respiratory syndrome coronavirus 2 (SARS-CoV-2) receptor-binding domain (anti-RBD) in stored antenatal serum samples over time between COVID-19 pre-pandemic and pandemic periods. The authors reported a higher prevalence of seroreactivity to SARS-CoV-2 on Sept 2019 compared to mid-Feb 2020. Overall it is an interesting study representing a retrospective analysis of the prevalence of seroreactivity over time and its association with variables. My comments to the authors are:

1. In the method, in the logistic regression, have you controlled potential confounders (such as BMI, previous history of flu infections, flu vaccination status etc.) potentially affected the outcome variable in the analysis?

No, data used were unlinked and anonymous: we were only able to include data relating to deprivation, age, ethnicity and sample-period only. Please see Methods, line 100-106 for further detail on included data:

‘Prior to anonymization, samples were cross-classified by fortnight, age group (18-29 years, 30-34 years, 35-44 years), ethnicity (all white, all Asian, all Black, Other), and deprivation (most-deprived two IMD deciles 1-2, intermediate four deciles 3-6, least-deprived four deciles 7-10) ... We gave an undertaking not to report on any cross-classification for which the observed count was under 25. This undertaking was met by pooling across adjacent fortnights. ‘

2. In the method ‘Defining seroreactivity’ section, please provide rationale and/or reference why the binding ratio (BR) ≥ 1 was considered as seroreactive for SARS-CoV-2 anti-RBD.

Thank you, we describe our approach in using the standard formula to determine an ELISA cut-off in the, ‘Defining Seroreactivity section,’ lines 144-150:

‘Defining seroreactivity

As set out in Rosadas et al.17, the cutoff for seroreactivity was established by adding 0.1 to the average of optical density (OD) obtained for three negative controls assayed in each run. The signal-to-cutoff value, known as the binding ratio (BR), for each sample was determined by dividing the sample OD by the cutoff OD. A sample was generally considered seroreactive for SARS-CoV-2 anti-RBD when the BR was greater than or equal to 1. BRs between 0.8 and 1.2 were considered to display a weak signal (borderline non-seroreactive; borderline seroreactive).’

3. In the result, table 3, is it adjusted or unadjusted odds?

The odds in table 3 are adjusted as based derived from regression analysis which legend states. Table 3 column title amended to read: 

‘Adjusted Odds Ratio.’

4. Some minor errors in the use of English throughout, i.e, missing/incorrect use of articles and prepositions - further proofreading required.

Thank you, I have re-read and I can’t see any further errors. I may be insensitive to my own mistakes. 

Reviewer 2

First of all, I would like to thank all authors.... this manuscript titled “Tracking the incidence and risk factors for SARS-CoV-2 infection using historical maternal booking serum samples" is easy to follow and well written. However, I have only one comment:

Could you please follow PLOS ONE guidelines, such as line number and reference style?

Thank you, formatting adjusted in line with PLOS ONE guidance. 

Reviewer 3 comment

1. Would like to hear more about the choice of sample size prior to starting the study. Maybe mention the possible limitations of generalizability of maternal cohorts when pregnant women may be more susceptible to infection.

Thank you. With regard to sample size, as noted in text, prior estimates based on population data on ethnicity did not fully reflect the ethnicity distribution for pregnant women but the chosen age-bands worked well. After high prevalence was attained, sample size was reduced to enable testing of samples further back in 2019. As described in lines 91-94 we had planned to analyse all available stored maternal serum samples from 4 NW London maternity centres from the period December 2019-May 2020. When seroreactive samples were found in the earliest time period and sufficiently narrow confidence were obtained with testing of 500 samples per fortnight, we adjusted our strategy as described in lines 107-116 to cover a wider time period in order to firstly test earlier samples to examine the onset of community transmission and secondly to test later samples to overlap with national serosurveillance studies for comparative purposes. This study was as pilot study to establish the testing period for a subsequent national study. 

With regards to generalisability and susceptibility, this is a good point and we have included the following in the discussion, lines 369-374:

‘Pregnant women’s increased susceptibility for some respiratory viruses make them a crucial group for surveillance, however the evidence for increased susceptibility to SARS-CoV-2 in pregnancy is mixed 31. Advice for pregnant women to shield may affect the generalisability of our data to other populations, however these samples were taken at around 10-12 weeks gestation when the majority of women would have only recently have become aware of pregnancy.’

---

## [Editor Report · Decision Letter 1]

19 Aug 2022

Tracking the incidence and risk factors for SARS-CoV-2 infection using historical maternal booking serum samples

PONE-D-22-11657R1

Dear Dr. Edward W. S. Mullins

We’re pleased to inform you that your manuscript has been judged scientifically suitable for publication and will be formally accepted for publication once it meets all outstanding technical requirements.

Kind regards,

Atalay Goshu Muluneh, MPH

Academic Editor

PLOS ONE
---

## [Editor Report · Acceptance letter]

24 Aug 2022

PONE-D-22-11657R1 

Tracking the incidence and risk factors for SARS-CoV-2 infection using historical maternal booking serum samples 

Dear Dr. Mullins:

I'm pleased to inform you that your manuscript has been deemed suitable for publication in PLOS ONE. Congratulations! Your manuscript is now with our production department. 

Kind regards, 

on behalf of

Mr. Atalay Goshu Muluneh 

Academic Editor

PLOS ONE